# Meloxicam and Dexamethasone Administration as Anti-Inflammatory Compounds to Sows Prior to Farrowing Does Not Improve Lactation Performance

**DOI:** 10.3390/ani11082414

**Published:** 2021-08-17

**Authors:** Kate J. Plush, John R. Pluske, David S. Lines, Cameron R. Ralph, Roy N. Kirkwood

**Affiliations:** 1SunPork Group, 1/6 Eagleview Place, Eagle Farm, QLD 4009, Australia; david.lines@sunporkfarms.com.au; 2Agricultural Science, College of Science, Health Engineering and Education, Murdoch University, Murdoch, WA 6150, Australia; j.pluske@april.org.au; 3Livestock Farming Systems Alliance, South Australian Research and Development Institute, Roseworthy, SA 5371, Australia; cameron.ralph@foodagility.com; 4School of Animal and Veterinary Science, The University of Adelaide, Roseworthy, SA 5371, Australia; roy.kirkwood@adelaide.edu.au

**Keywords:** sow, inflammation, survival, growth, welfare, reproduction

## Abstract

**Simple Summary:**

Sows may experience pain and discomfort whilst giving birth. Additionally, the birthing process is accompanied by an inflammation response. Administering anti-inflammatory compounds prior to birth may provide an opportunity to improve piglet survival and growth. The aim of this experiment was to assess the efficacy of both a non-steroidal (NSAID; meloxicam) and steroidal (SAID; dexamethasone) anti-inflammatory drug for improving farrowing house performance of sows. In younger sows, there was no impact of treatment; however, older sows from the NSAID treatment gave birth to fewer live piglets. Postnatal mortality was unaffected by treatment and no improvement in piglet growth was observed. Feed intake of both NSAID and SAID sows was improved when compared with the control group; however, there was a tendency for a delayed oestrus in the NSAID group. Administering NSAID to sows prior to farrowing is not recommended as it reduces piglet survival and subsequent reproduction.

**Abstract:**

The aim of this experiment was to determine whether administration of an anti-inflammatory compound to sows prior to farrowing would, via reduced pain and inflammation, increase piglet survival and growth. At day 114 of gestation, multiparous sows were randomly allocated to one of the following treatments: Control (*n* = 43), which received 10 mL saline, NSAID (*n* = 55) which received 0.4 mg/kg meloxicam and SAID (*n* = 54) which received 0.1 mg/kg dexamethasone. Treatments were applied again on day 116 if farrowing had not occurred. There was no treatment effect on piglets born alive or dead from parity two to four sows but in those of parity five and older, NSAID administration reduced the number of piglets born alive and increased the number of piglets born dead (*p* < 0.05). Sow rectal temperature and incidence of mastitis were unaffected by treatment (*p* > 0.05). Lactation day two plasma concentrations of cortisol, prostaglandin F2 alpha metabolite and haptoglobin did not differ among treatments (*p* > 0.05). Treatment effects were not observed in liveborn piglet mortality at any age, or litter weight at day 21 (*p* > 0.05). Average feed intake during lactation was increased by both NSAID and SAID treatments (*p* = 0.001). The use of meloxicam prior to farrowing should be avoided as it reduced the number of piglets born alive and did not improve piglet survival and growth.

## 1. Introduction

There are life history stages where pain is unavoidable, and parturition is one of these times. Modern sows have been heavily selected for reproductive traits that have increased litter size, duration of farrowing, and consequently the associated pain [1]. The way in which we house sows may act to amplify this pain. Prior to parturition, the crated environment in which sows are housed has been suggested to be uncomfortable and so may result in soreness and injury [2]. In the initial stages of parturition, confinement within a crate prohibits nest-building behaviour, reducing circulating oxytocin levels [3]. In rats, there is evidence that central oxytocin has analgesic properties [4], thus crated sows may experience amplified levels of discomfort, and behavioural indicators would support this notion [5]. During the expulsive phase, pain is associated with the release of opioids [6] that inhibit oxytocin release [7]. Treatment with an opioid antagonist shortens parturition, while increased pain during parturition increases sow movements [8], the latter being potentially dangerous to neonatal piglets. If providing sows with pain relief prior to parturition accelerates the farrowing process, the provision of an anti-inflammatory compound may improve sow wellbeing and piglet survival.

Anti-inflammatory compounds are classified as either non-steroidal anti-inflammatory drugs (NSAIDs) or steroidal anti-inflammatory drugs (SAIDs) [9]. The NSAIDs act on cyclo-oxygenase enzymes (COX-1 and/or COX-2), inhibiting conversion of membrane-associated arachidonic acid to eicosanoids (e.g., prostaglandins). NSAIDs include meloxicam, which acts on both COX-1 and 2 with a half-life of 6 to 7 h. Glucocorticoids enter cells, attach to their receptors and regulate gene expression for anti-inflammatory proteins such as lipocortin-1, which inhibits phospholipase A-2, so preventing production of arachidonic acid. SAIDs include the high potency corticosteroid, dexamethasone, whose duration of action is approximately 48 h. In cattle, recent studies have demonstrated that even in healthy cows, inflammatory mediators are elevated following parturition [10]. Further, NSAID treatments in the immediate post-partum period can increase milk yield [11].

In pigs, Tenbergen et al. [12] reported no improvement in piglet average daily gain when the NSAID meloxicam was administered to sows 12 h post-partum. In contrast, Mainau et al. [13] demonstrated that an oral dose of meloxicam to the sow at the onset of parturition improved piglet plasma IgG concentrations and average daily gain to weaning. The impacts of glucocorticoids (e.g., dexamethasone) remain to be elucidated, but there is preliminary evidence of improved piglet growth [14]. Therefore, the aim of the present experiment was to determine the effectiveness of NSAID (meloxicam) and SAID (dexamethasone) treatments prior to farrowing on the farrowing and lactation performance of sows and, by extension, the performance of their piglets. It was hypothesised that their administration prior to farrowing would benefit piglet survival and growth.

## 2. Materials and Methods

### 2.1. Animals

This experiment was conducted at a South Australian 7500-sow commercial breeder unit during summer months in 2018 after approval from the PIRSA Animal Ethics Committee (approval # 26-17). One hundred and fifty-two multiparous sows entered farrowing accommodations 4.9 ± 0.2 days prior to farrowing. No sows were induced to farrow and farrowing occurred over an eight-day period with an average gestation length of 116.6 ± 0.2 days. Sows were fed 2.5 kg/d of lactation sow mash formulated to provide 14.25 DE MJ/kg and 0.087% SID lysine from entry to the farrowing shed until the day of farrowing and, thereafter, fed ad libitum until weaning (hoppers held ~7.5 kg of feed and feed was delivered twice daily). The date of farrowing was recorded once daily between 07:00 h and 10:00 h, and piglet fostering occurred after this event. Fostering was minimal and piglet movement only occurred when the number of piglets exceeded the number of functional teats available. In this event, split suckling was carried out prior to piglet relocation according to the procedures of Huser et al. [15] to ensure more equal colostrum intake across the litter. After the initial fostering event, which occurred within treatment where possible, piglets were only moved from the sow if they were losing condition, and at this point, they were taken off experiment and recorded as ‘ill thrift’. The litters were weaned at 25.4 ± 0.1 days and this lactation length was consistent across treatment groups. At weaning, sows were vaccinated against leptospirosis, parvovirus and erysipelas and were relocated to a breeding barn and housed in groups of 40. Starting on day three post weaning, sows were run in front of mature boars for oestrus detection. At detection and again 24 h later, sows were artificially inseminated with 3 × 10^9^ sperm in 80 mL diluent supplied by a commercial boar stud.

### 2.2. Housing

Sows were housed in conventional farrowing crates, having fully slatted plastic flooring and provided the sow with a feed hopper, two nipple drinker lines (one for standing and one for lying access), and a triangular creep area with solid flooring and heat lamp for the piglets. The farrowing shed contained 237 identical crates placed in three rows and was naturally ventilated. The average weather conditions experienced during farrowing were a minimum temperature of 11.3 °C, maximum temperature of 29.1 °C, and relative humidity of 51.5%. The lighting regime followed daylight hours (06:00 h to 20:00 h).

### 2.3. Treatments

At day 114 of gestation, healthy sows were randomly allocated to one of three treatments. Control sows received 10 mL saline, SAID sows 0.1 mg/kg dexamethasone (2 mg/mL Dexason, Troy Laboratories, Glendenning Australia, and NSAID sows 0.4 mg/kg meloxicam (20 mg/mL Recocam, Abby Animal Health Pty Ltd, Glendenning Australia). All sows received treatment via intramuscular injection. The Control treatment consisted of, *n* = 33 parity 2 to 4 sows, and *n* = 10 parity 5+; NSAID, *n* = 39 parity 2 to 4, and *n* = 16 parity 5+; and SAID, *n* = 40 parity 2 to 4, and *n* = 14 parity 5+. Sows were treated at 15:00 h on day 114 and again on day 116 if they had failed to farrow (15 Control, 22 NSAID and 33 SAID sows). After treatment, all sows were managed similarly.

### 2.4. Measurements

On the day of farrowing, sow location, parity, and litter details (total born, born alive, and born dead) were noted. A four-point scoring system was developed to quantify fresh sow injury around farrowing based on three facial regions (nose, snout, and eyes/ears); (0) no abrasions evident, (1) abrasions were localised to one of three facial zones, (2) abrasions were localised to two of the three facial zones, and (3) abrasions were present at all three facial zones. Rectal temperatures of each sow were recorded at 14:00 h on days one, two, and three after farrowing, and incidences of temperatures greater than 40 °C were noted. If a sow presented with presumptive mastitis or endometritis, they were medicated with penicillin as per veterinary instruction, but did not receive any further anti-inflammatory medication (five sows on day one, nine sows on day two, and five sows on day three). These sows were removed from all measurements collected after farrowing and so piglet mortality and growth was collected from *n* = 35 CON sows, *n* = 51 NSAID sows, and *n* = 47 SAID sows.

A 10 mL blood sample was collected on day two of lactation from 20 sows (*n* = 15 parity 2 to 4 and *n* = 5 parity 5+ sows) per treatment via jugular venepuncture using an 18 g needle and heparinised vacutainer, after restraint by snare. This sample was stored on ice until centrifugation and plasma stored frozen in duplicate at −20 °C. At this same time, three piglets of average weight were selected from this subset of sows, and blood was collected via jugular venepuncture using a 23 g needle, 5 mL syringe and serum tube. The sample was refrigerated overnight and then assessed for protein percentage using Brix refractometry for a crude indication of colostrum intake [16].

The number of piglets present in the litters and the litter weights were recorded after fostering at day one and again at day 21. All piglet mortalities were recorded. Sows were fed twice daily at 07:00 h and 16:00 h by hand using a feed cart that had been converted to scales. Due to the design of the feeders (Crystal Spring wet/dry sow feeder, Stockyard Industries, North Bendigo Australia), any feed wastage was retained in the bowl below the hopper, and before each feeding event, this wastage was returned to the hopper. Each feeder was filled to a standardised volume (marked by a lip at the top of the hopper), and so the feed delivered on each day was summed to give a daily feed intake for each sow.

### 2.5. Laboratory Analysis

Sow plasma samples were analysed for cortisol, 13,14-dihydro-15-keto-prostaglandin F2-alpha metabolite (PGFM), and haptoglobin concentrations as markers of stress and inflammation. Cortisol concentration was determined by radioimmunoassay (#07221102, MP Biomedicals, NSW Australia) with average intra- and inter-assay coefficients of variation (CV) of 25% and 6.7%, respectively. The PGFM analyses was conducted using an ELISA (#MBS7214882, Resolving Images, VIC Australia) with average intra- and inter-assay CV of 12% and 20%, respectively. Haptoglobin analyses were conducted using an ELISA (#ab205091, Abcam, VIC, Australia) with intra- and inter-assay CVs of 2.6% and 6.0%, respectively.

### 2.6. Statistics

Using data obtained from the experimental site previously (*n* = 989, mean = 59.1, standard deviation = 14.7 and variance = 0.25) a power calculation was conducted prior to the application of treatments using day 21 litter weight as the main variable. It was determined that with *n* = 50 sows per treatment, we would be able to detect a difference in day 21 litter weight of 5.7 kg. All data were analysed in SPSS Statistics V24 (IBM Corporation, Armonk, NY, USA). Unless otherwise specified, a general linear model was applied with the fixed effects of room (one to four), parity group (2 to 4, or 5+), treatment (CON, SAID or NSAID), and the interaction between parity and treatment. Facial injury score and piglet deaths (born dead, pre-foster mortality, post-foster mortality, liveborn mortality, and total mortality) were analysed using a generalised linear model with Poisson distribution and the same model. The number of piglets born dead also contained the covariate of total born. Sow feed intake was analysed using a linear mixed model with sow as the subject and day as the repeated measure. Parity group, treatment, day (1 to 21) and the interactions parity group by day, treatment by day, and parity group by treatment by day were included in the model as fixed effects, and room as a random effect. Significance was established when *p* < 0.05, tendency at *p* < 0.10 and Bonferroni post hoc tests were carried out to determine significant comparisons. Data presented are mean ± standard error of the mean.

## 3. Results

The average parity of sows was 3.8 ± 0.1 and gestation length 116.5 ± 0.3 days, and both were unaffected by treatment. There were no overall treatment effects on the total number of piglets born within a litter (13.4 ± 0.6), or the number of piglets born dead (0.8 ± 0.1), but there was a tendency for the number of piglets born alive to be reduced in the NSAID treatment (11.4 ± 0.5 versus Control 12.9 ± 0.6 and SAID 12.7 ± 0.5; *p* = 0.10). The treatment by parity interaction was significant for both piglets born alive and stillbirths. NSAID-treated sows in the parity 5+ group gave birth to fewer liveborn and more stillborn piglets (Figure 1; *p* < 0.05).

The main effect of treatment did not affect facial injury score (*p* > 0.05). However, within parities 2 to 4, SAID sows presented with a lower score than Control, with NSAID sows intermediate (Figure 2; *p* < 0.05).

There was no treatment effect on the rectal temperature of sows over the first 3 days after farrowing. Similarly, no treatment effect on incidence of mastitis was evident during this period (Table 1). There was a tendency for piglets born to sows from both the NSAID and SAID treatments to record lower serum protein concentrations at 24 h of age when compared to those born from Control sows (*p* = 0.10).

Treatment had no impact on sow plasma cortisol or PGFM concentrations on day 2 (Table 2; *p* > 0.05). Parity tended (*p* = 0.10) to influence cortisol concentration, with parity 5+ sows displaying increased concentrations compared to those of parity 2 to 4, whilst PGFM was reduced in 5+ sows when compared to younger sows (*p* < 0.05). Treatment had no impact on haptoglobin concentration but there was a trend for reduced levels in older sows (*p* < 0.10).

There was no impact of treatment on the total number of piglet deaths or deaths from sow overlay or low piglet viability on days one or two (*p* > 0.05). Similarly, piglet mortality across all stages or piglet removal failed to be influenced by treatment (Figure 3; *p* > 0.05). Whilst most stages of mortality remained unaffected by parity, total deaths (the sum of piglets born dead and postnatal mortality) were increased in parity 5+ sows (2.8 ± 0.3 piglets) when compared with parity 2 to 4 sows (2.1 ± 0.1 piglets; *p* < 0.05).

The number and weight of piglets at day one and day 21 of age were unaffected by treatment (Table 3; *p* > 0.05). Parity did not influence any of the measures at day one, but litter size at day 21 was reduced in the 5+ sows compared with the 2 to 4 sows (9.3 ± 0.3 versus 9.8 ± 0.2 pigs, respectively; *p* < 0.05). A similar pattern was observed for the day 21 litter weight (parity 5+: 58.5 ± 2.0 kg, and parity 2 to 4: 64.7 ± 1.2 kg; *p* < 0.05). Feed intake was affected by day (*p* < 0.001) and was higher for NSAID and SAID treated sows than controls (Table 3; *p* = 0.001), but the interaction between day and treatment was not significant (*p* > 0.05). Sows of parity 2 to 4 displayed a higher average daily feed intake (7.6 ± 0.1 kg) than those of parity 5+ (7.0 ± 0.1 kg; *p* < 0.001).

After weaning, there was a tendency for treated sows to return to oestrus sooner, with SAID sows exhibiting a shorted wean to service interval (*p* = 0.10). Litter sizes were not affected by treatment (Table 4).

## 4. Discussion

Older parity sows treated with the NSAID gave birth to more stillborn and, consequently, fewer liveborn piglets. Anti-inflammatory drugs act by inhibiting prostaglandin synthesis [17], so the finding that stillbirths were increased in older parity sows from this treatment is not surprising. Prostaglandins are pivotal for the contraction of smooth muscle and hence for the uterus to expel the foetus during birth. If prostaglandin concentrations were reduced in this tissue by the NSAID treatment, this could have delayed the birth of piglets and resulted in the observed increase in intra-partum death. Although speculative, an alternative explanation could be, as this was most evident in the older sows, a clinically unapparent hypocalcaemia due to repeated lactations accentuating the anti-prostaglandin effect on uterine contractions.

That a similar effect was not evident for SAID-treated sows implies multiple modes of action. As reviewed by Antonucci et al. [18], in human medicine NSAIDs are not given to women in late gestation because of possible effects on the unborn child. This includes an adverse effect on the ductus arteriosus whose patency is maintained by prostaglandins. An associated depressed prostaglandin level could impact cardiovascular health in addition to adverse effects on brain, kidney, lung, and gastrointestinal tract [18]. It is reasonable to assume the same is true for pigs with the possible consequence of more stillbirths. Corticosteroids also depress prostaglandins, but an effect of the SAID treatment on stillbirths was not apparent in the present study. An explanation for this may involve the known increase in pre-partum corticosteroid binding globulin (CBG) [19]. The administered dexamethasone is more potent than endogenous corticoids, but foetal blood levels would be expected to be lower with some bound to CGB, so limiting effects on prostaglandin synthesis.

Interestingly, piglet serum protein levels, as a measure of colostrum transfer, tended to be lower in the NSAID piglets, which contrasts with previous results [13]. However, the latter investigation administered the NSAID during parturition while in the current study it was administered pre-partum. The average duration from administration to farrowing was 1.5 days (data not shown). Indeed, Ward et al. [14] has recently demonstrated that administering an anti-inflammatory too far out from farrowing results in adverse outcomes for the piglet. Given that the increase in stillbirths in the NSAID treatment is indicative of farrowing difficulties, then likely the liveborn piglets from this treatment would have experienced birth trauma/hypoxia. Birth hypoxia has been linked to poor piglet vitality [20]. So, this poorer vitality, resultant from farrowing difficulties, may explain the reduced serum protein levels in the NSAID litters. Future work in this area should include behavioural observations on both the sow and piglets to confirm whether farrowing duration and piglet vigour are impacted by anti-inflammatory administration.

Circulating PGFM levels reflect PGF2α biosynthesis and so were used as a measure of prostaglandin-mediated inflammation [21]. However, a failure to detect any treatment difference may reflect the short (15 min) half-life of PGFM and the need for more intensive sampling. Prostaglandin F2α is integral to the inflammatory process, and inflammation also induces acute phase proteins, including haptoglobin and C-reactive protein (CRP). Haptoglobin is commonly cited as being elevated during the transition from gestation to lactation in dairy cows [22], but was also unaffected by treatment in the current experiment. Anti-inflammatory agents have been shown to be successful in preventing the elevation of this acute-phase protein [23]. Interestingly, while serum levels of haptoglobin and CRP have been shown to have minimal utility as an indicator of peripartal disease [24], haptoglobin is transferred to piglets via colostrum, and colostrum concentrations were negatively associated with litter growth rate [25]. Similarly, but to a lesser extent, piglet serum haptoglobin concentrations were negatively associated with growth [25]. The absence of any treatment effect on acute-phase proteins in the present study was unexpected. The only logical explanation for this could be the delay between treatment administration and blood sampling for haptaglobin. As sows were treated on day 114 of gestation, with the reported average gestation length being 116 days, and blood sampling occurring at day two of lactation, perhaps the anti-inflammatory action of the medications tested was exhausted. Future work should test their use in combination with parturition induction.

Sow wellbeing was measured indirectly by a number of variables as the premise for administering the anti-inflammatory compounds before farrowing was to improve sow comfort during parturition. We have recently constructed a facial injury score that quantifies fresh injuries present on the nose, snout, and eyes/ears of a sow immediately following parturition in a similar fashion to what has been used previously for piglets [26] and sows [27] in aggression investigations. We hypothesise that a higher score is a result of hard crate fixtures inflicting injuries to the face of the sow during redirected nesting [28], and movements induced by a painful parturition [29]. Whilst validation of the score is currently underway (Plush et.al., 2021; in press), the fact the younger SAID treated sows had lower levels of injuries would suggest their wellbeing was improved. A longer farrowing duration is thought to negatively impact sow welfare due to increased pain and risk of dystocia [30]. We were unable to directly measure farrowing duration in the present study as the experiment was conducted on a commercial farm, but given the link between farrowing duration and intra-partum death [31], piglets born dead was used as an indicator of farrowing ease. Given that a higher number of piglets were born dead in the older, NSAID treated sows, it could be argued that wellbeing was reduced in this population. There was no treatment impact on the measured physiological indicator of wellbeing, as circulating cortisol concentration was not altered on day two by anti-inflammatory administration. Finally, treatment with a SAID or NSAID resulted in increased sow feed intake, which may also indirectly indicate an improved sow wellbeing. In support, the feed intake effect was most pronounced during days two to five after farrowing (data not shown), possibly indicating an improved post-partum recovery. The feed intake data support earlier work where sows lost less body condition and presented with fewer shoulder sores after ketoprofen administration [32].

Despite this improvement in feed intake by the NSAID and SAID treated sows, litter weight at weaning was not increased by either treatment. Thus, no relationship between sow feed intakes and pre-weaning piglet growth were detected in the current study. This suggests that within the feed intakes observed, milk yield was not a compensable effect, i.e., milk yields will decrease with very low feed intakes but, beyond a certain point, will not increase with increasing feed intake. Additionally, no improvement in post-natal piglet survival was noted, and this is perhaps not surprising given the lack of consistent treatment impacts on sow wellbeing and inflammatory markers. There was some evidence that sow parity may interact with the treatments imposed in the current study, but this requires validation across a larger population.

Data from the present experiment indicate a negative effect of anti-inflammatory drugs on subsequent reproductive performance, namely the wean to service interval. It is possible that inhibition of prostaglandins impaired early uterine involution with adverse consequences for uterine health. This is supported by the observation that prostaglandin analogue injected 24–48 h post farrowing increased the next litter size in older parity sows [33]. There may also be an interaction between the treatments imposed and farrowing hygiene. The inflammatory response that is induced during the transition phase improves the clearance of pathogens and so risk of infection post-partum [22], which is high. Reduced reproduction and low farrowing hygiene are linked in sow herds [34], and so perhaps the combination of increased disease loading and decreased inflammatory response also contributed to this poorer reproductive performance of the NSAID-treated sows. However, why this was also not observed in the SAID sows is unclear.

## 5. Conclusions

In conclusion, meloxicam should not be administered prior to farrowing as there is an increased risk of intra-partum piglet death, poorer neonatal piglet serum protein levels, and impaired subsequent reproduction. There was some suggestion that dexamethasone improved sow wellbeing, as indicated by reduced level of injuries around farrowing and improved feed intake, but pre-partum steroidal anti-inflammatory injection did not improve postnatal piglet survival or growth.

## Figures and Tables

**Figure 1 animals-11-02414-f001:**
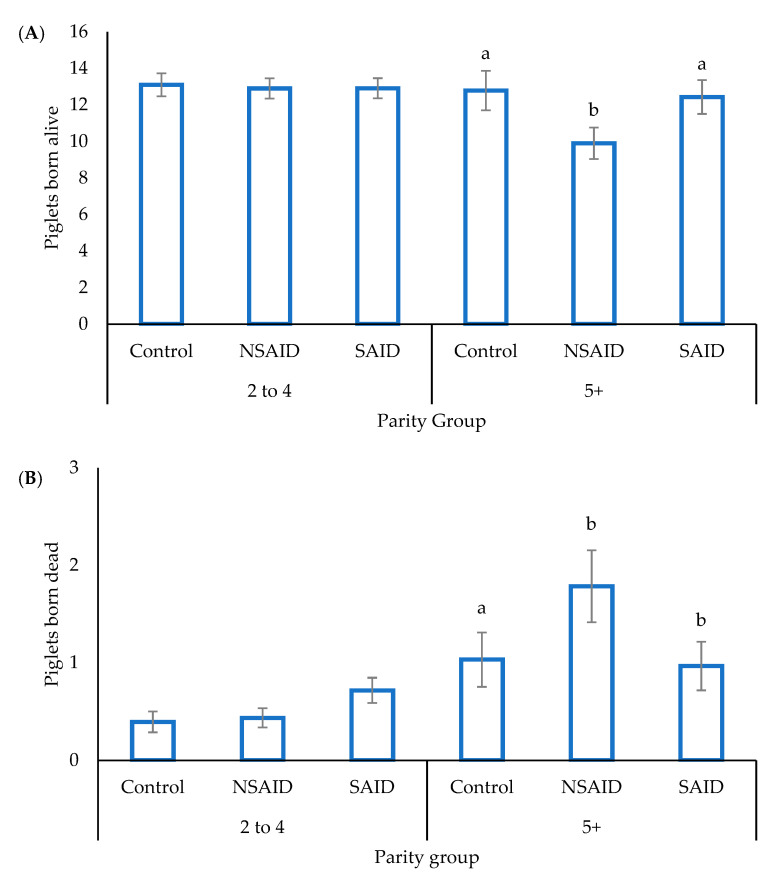
The effect of saline (Control), a non-steroidal anti-inflammatory (NSAID), or steroidal anti-inflammatory (SAID) administration prior to farrowing on the number of piglets born alive (**A**) and dead (**B**) per litter (mean ± SEM) for sows of parity 2 to 4 or 5+. a,b represents a significant difference (*p* < 0.05) within parity group.

**Figure 2 animals-11-02414-f002:**
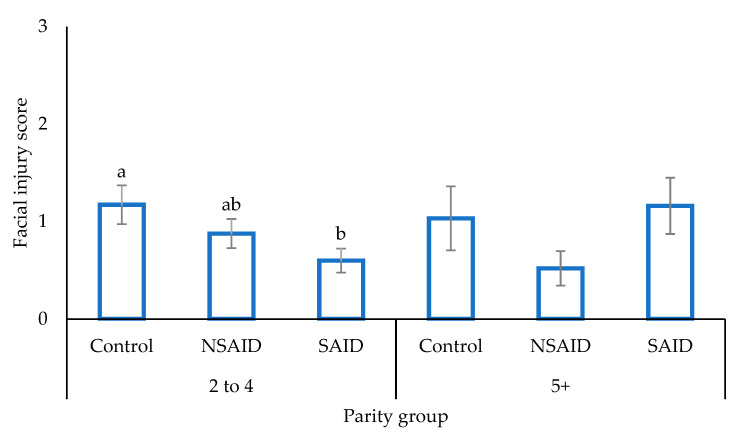
The mean ± SEM facial injury score (with 0 being no injury, and 3 being a high level of injury) for sows of parity 2 to 4 or 5+ injected with either saline (Control), a non-steroidal anti-inflammatory (NSAID), or a steroidal anti-inflammatory (SAID) prior to farrowing. a,b represents a significant difference (*p* < 0.05) within parity group.

**Figure 3 animals-11-02414-f003:**
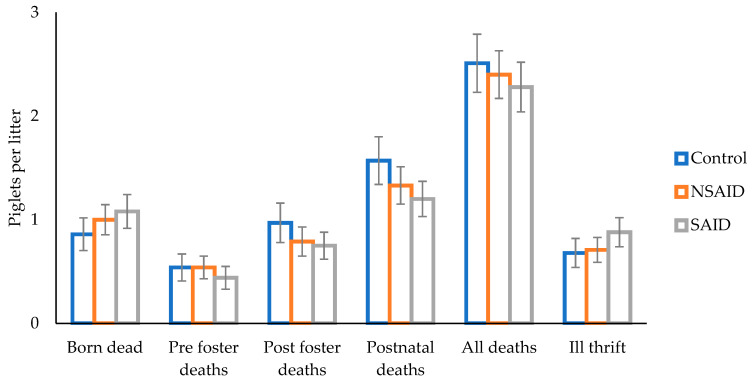
The effect of saline (Control), a non-steroidal anti-inflammatory (NSAID), or steroidal anti-inflammatory (SAID) administration prior to farrowing on piglet deaths recorded at birth, prior to fostering, after fostering, postnatal deaths (the sum of pre and post foster deaths), all piglet deaths (the sum of piglets born dead and postnatal deaths), and piglet removal for ill thrift from sows (mean ± SEM).

**Table 1 animals-11-02414-t001:** The effect of saline (Control), a non-steroidal anti-inflammatory (NSAID), or steroidal anti-inflammatory (SAID) administration prior to farrowing on sow rectal temperature recordeddaily for three days post farrowing, the incidence (%) of mastitis (> 40 °C) over the 3 day recording period, and Brix piglet serum protein (%) at 24 h of age (mean ± SEM). ^†^ 95% Confident intervals presented rather than SEM for binary data.

		Control		NSAID		SAID		*p*-Value
		Mean	SEM	Mean	SEM	Mean	SEM	
Rectal temperature (°C)	Day 1	38.9	0.1	38.8	0.1	39.0	0.1	0.349
	Day 2	38.9	0.1	38.9	0.1	38.7	0.1	0.254
	Day 3	38.8	0.1	38.7	0.1	38.8	0.1	0.444
Incidence of mastitis (%) ^†^		19	(9–35)	7	(2–17)	12	(6–25)	0.211
Brix piglet serum protein (%)		6.5	0.4	5.5	0.4	5.7	0.4	0.100

**Table 2 animals-11-02414-t002:** Day 2 plasma square root (sqrt) cortisol, PGFM and haptoglobin concentrations in sows injected with either saline (Control), a nonsteroidal anti-inflammatory (NSAID), or steroidal anti-inflammatory (SAID) prior to farrowing, and those of parity 2 to 4 and 5+. ^†^ Back-transformed means are presented in brackets.

	Treatment						*p*-Value	Parity				*p*-Value
	Control		NSAID		SAID			2 to 4		5+		
	Mean	SEM	Mean	SEM	Mean	SEM		Mean	SEM	Mean	SEM	
Sqrt cortisol (nmol/l) ^†^	7.9	0.8	8.7	1.0	6.3	0.9	0.176	6.8	0.6	8.5	0.9	0.100
	(62.4)		(75.7)		(39.7)			(46.2)		(72.3)		
PGFM (ng/mL)	4.7	0.6	3.2	0.7	4.0	0.6	0.271	4.8	0.4	3.1	0.6	0.023
Haptoglobin (mg/mL)	1.4	0.1	1.4	0.2	1.2	0.1	0.387	1.5	0.1	1.2	0.1	0.058

**Table 3 animals-11-02414-t003:** The effect of saline (Control), a non-steroidal anti-inflammatory (NSAID), or steroidal anti-inflammatory (SAID) administration prior to farrowing on litter size, litter and average piglet weight measured on day one and day 21, and average lactation feed intake (mean ± SEM). ^a,b^ represents a significant difference (*p* < 0.05) within measure.

	Control		NSAID		SAID		*p*-Value
	Mean	SEM	Mean	SEM	Mean	SEM	
Litter size at fostering	11.2	0.2	11.4	0.2	11.3	0.2	0.748
Litter weight day 1 (kg)	14.8	0.6	15.6	0.5	15.9	0.5	0.358
Litter size day 21	9.6	0.3	9.6	0.3	9.7	0.3	0.981
Litter weight day 21 (kg)	60.5	2.3	61.4	1.9	61.3	2.1	0.952
Average sow feed intake per day (kg)	7.0 ^a^	0.1	7.4 ^b^	0.1	7.5 ^b^	0.1	0.001

**Table 4 animals-11-02414-t004:** Effects on subsequent reproductive performance for sows injected with saline (Control), a nonsteroidal anti-inflammatory (NSAID) or steroidal anti-inflammatory (SAID) prior to the previous farrowing event; means ± SEM.

	Control		NSAID		SAID		*p*-Value
	Mean	SEM	Mean	SEM	Mean	SEM	
Wean to service interval (days)	9.8	1.5	7.1	1.1	4.9	2.2	0.100
Next litter size	13.7	0.8	14.2	10.7	12.6	0.7	0.221

## Data Availability

The data presented in this study are available on request from the corresponding author.

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
