# Peer review of "Meloxicam and Dexamethasone Administration as Anti-Inflammatory Compounds to Sows Prior to Farrowing Does Not Improve Lactation Performance"

_animals, 2021, doi:10.3390/ani11082414_

Round 1

Reviewer 1 Report

Introduction:

  1. The hypothesis is very unspecific and does not match the measurements. The authors hypothize that the treatments would improve sow comfort but "sow comfort" was not investigated through the whole experiment. Additionally the authors responded that "sow comfort" is an assumption, but according to the GSP assumptions should not be part of the hypothesis. therefore the hypothesis should be revised.
  2. L 179ff: Also the Term "reducing the inflammation associated with parturition" is very unspecific and should be formulated more precisely.

Material and Methods:

  1. L 305: add "n=" to 5 parity 5+sows
  2. L 372: add the average weight of the piglets (average value +-SD)
  3. L 383: the subheading should get an own line
  4. L 292 ff: the number of sows of the Control and SAID group does not match with the number of sows from the abstract.
  5. L 302 ff: a total of 13 sows were removed because of endometritis. I suggest to mention from with groups these sows are and write down the final number of used sows in each group. otherwise, the n/group from L 292 ff is misleading.

Statistics:

I would suggest to add a short description of the power test analysis.

Conclusions:

L 829: the term "wellbeing" is associated with more factors than just "feed intake". given the fact that sow wellbeing was not measured in the present study I would strongly suggest not using this term in this context.

L 830: the authors wrote that anti-inflammatory injection "did little" to improe postnatal piglet survival. I would suggest du rewrite this sentence since in L 521ff the authors wrote that there was no treatment effect on number of piglets or weight gain.

Author Response

  1. The hypothesis is very unspecific and does not match the measurements. The authors hypothize that the treatments would improve sow comfort but "sow comfort" was not investigated through the whole experiment. Additionally the authors responded that "sow comfort" is an assumption, but according to the GSP assumptions should not be part of the hypothesis. therefore the hypothesis should be revised.

The authors agree and so have made the hypothesis specific. It now read:

It was hypothesised that their administration prior to farrowing would benefit piglet survival and growth.

  1. L 179ff: Also the Term "reducing the inflammation associated with parturition" is very unspecific and should be formulated more precisely.

This mention has now been removed from the hypothesis.

Material and Methods:

  1. L 305: add "n=" to 5 parity 5+sows

The authors apologise, but the line numbers do not match up with the copy of the manuscript downloaded. We have changed the following, and hope this addresses the reviewers concern:

The Control treatment consisted of, n=33 parity 2 to 4 sows, and n=10 parity 5+, NSAID; n=39 parity 2 to 4, and n=16 parity 5+, and SAID n=40 parity 2 to 4, and n=14 parity 5+.

  1. L 372: add the average weight of the piglets (average value +-SD)

Again, line numbers do not match but if the reviewer is referring to the reference o average piglet weights at day 1 and 21 in the methods, we disagree as these are results. No other measure has been presented in this way (gestation length, litter size, etc) and so we have left the day 1 and 21 weights in the results section.

  1. L 383: the subheading should get an own line

Apologies, this was an editing error and has now been rectified (once track changes accepted).

  1. L 292 ff: the number of sows of the Control and SAID group does not match with the number of sows from the abstract.

We have gone back to the original dataset and confirmed the number of sows used in the statistical analyses:

  • Total = 152 sows
  • CON; n = 43 sows, NSAID; n = 55 sows and SAID; n = 54 sows

  1. L 302 ff: a total of 13 sows were removed because of endometritis. I suggest to mention from with groups these sows are and write down the final number of used sows in each group. otherwise, the n/group from L 292 ff is misleading.

This section has now been updated and reads:

‘’ If a sow presented with presumptive mastitis or endometritis, they were medicated with penicillin as per veterinary instruction but did not receive any further anti-inflammatory medication (five sows on day one, nine sows on day two and five sows on day three). These sows were removed from all measurements collected after farrowing and so piglet mortality and growth was collected from n= 35 CON sows, n=51 NSAID sows, and n=47 SAID sows.’’

Statistics:

I would suggest to add a short description of the power test analysis.

The beginning of the statistics section now reads:

“Using data obtained from the experimental site previously (n=989, mean=59.1, standard deviation=14.7 and variance=0.25) a power calculation was conducted prior to the application of treatments using day 21 litter weight as the main variable. It was determined that with n=50 sows per treatment, we would be able to detect a shift in day 21 litter weight of 5.7kg.’’

Conclusions:

L 829: the term "wellbeing" is associated with more factors than just "feed intake". given the fact that sow wellbeing was not measured in the present study I would strongly suggest not using this term in this context.

We thank the reviewer for their comment and fear we have not given enough discussion and evidence for the use of the word ‘’wellbeing’’. We did collect a facial injury score but removed this from the results section prior to submission. We have now included this score in the revision. We have also clarified a number of other measures, that when taken collectively, should justify our use of the word wellbeing. This extra paragraph now reads:

‘’Sow wellbeing was measured indirectly by a number of variables as the premise for administering the anti-inflammatory compounds before farrowing was to improve sow comfort during parturition. We have recently constructed a facial injury score that quantifies fresh injuries present on the nose, snout, and eyes/ears of a sow immediately following parturition in a similar fashion to what has been used previously for piglets [26] and sows [27] in aggression investigations. We hypothesise that a higher score is a result of hard crate fixtures inflicting injuries to the face of the sow during redirected nesting [28], and movements induced by a painful parturition [29]. Whilst validation of the score is currently underway (Plush et.al., 2021; in press), the fact the younger SAID treated sows had lower levels of injuries would suggest their wellbeing was improved. A longer farrowing duration is thought to negatively impact sow welfare due to increased pain and risk of dystocia [30]. We were unable to directly measure farrowing duration in the present study as the experiment was conducted on a commercial farm, but given the link between farrowing duration and intra-partum death [31], piglets born dead was used as an indicator of farrowing ease. Given that a higher number of piglets were born dead in the older, NSAID treated sows, it could be argued that wellbeing was reduced in this population. There was no treatment impact on the measured physiological indicator of wellbeing, as circulating cortisol concentration was not altered on day two by anti-inflammatory administration. Finally, treatment with a SAID or NSAID resulted in increased sow feed intake, which may also indirectly indicate an improved sow wellbeing. In support, the feed intake effect was most pronounced during days two to five after farrowing (data not shown), possibly indicating an improved post-partum recovery. The feed intake data support earlier work where sows lost less body condition and presented with fewer shoulder sores after ketoprofen administration [32].’’

L 830: the authors wrote that anti-inflammatory injection "did little" to improe postnatal piglet survival. I would suggest du rewrite this sentence since in L 521ff the authors wrote that there was no treatment effect on number of piglets or weight gain.

Agreed. The authors have changed ‘’did little to’’ to ‘’did not’’

Reviewer 2 Report

The amendments according to the reviwers' comments and recommendations are appreciated.

Author Response

The amendments according to the reviwers' comments and recommendations are appreciated.

We thank the reviewer for their time, and for strengthening our manuscript.

Round 2

Reviewer 1 Report

L28: "...dexamethason.." - One dot instead of two

L308: "...experiment. and anti-inflammatory..." -after a dot, a new sentences should not start with "and"

This manuscript is a resubmission of an earlier submission. The following is a list of the peer review reports and author responses from that submission.

Round 1

Reviewer 1 Report

The present study investigated the questions if meloxicam and dexamethasone given to sows prior to birth enhance piglets health and survial rate.

Introduction:

Overall, the introduction is focusing on elder references and some current references are missing. This should be added.

Line (L) 45-48: The first sentence is difficult to understand and makes no sense in the overall context of the introduction. This sentence should be rephrased or rewritten.

L 76 ff: The content of the stated aims of this study does not match the hypotheses that have been made. Aims are determining the effectiveness of Meloxicam and dexamethasone on farrowing and lactation performance of the sows and the hypothesis are focusing on sow´s comfort and reduced inflammatory processes and on piglets survival and growth. The authors should write this more precisely and decide on a focus.

L 80: please specify the term “reduce inflammation”, which inflammation is meant and where?

L 79ff: the authors hypothesied that the treatment improve sow comfort. There is no information in M+M how “sow comfort” was measured nor is there any information in the results section. This should be added, otherwise it must be assumed that some of the hypotheses have not been processed.

Materials and Methods:

Some additional information about the overall health status (e.g.PRRSV status) of the sow herd and vaccination programs (PRRSV, PCV2, Parvovirus, Erysipelas,…) would be very interesting. This information should be added.

L 123ff: A total of 13 sows developed mastitis or endometritis. Is there any information to which parity these sows belonged to? I would strongly recommand to exclude these sows from the study, as it is known that mastitis reduces the supply of milk/colostrum to the piglets and therefore effect their postnatal development. Additionally, antibiotics can be transmitted through milk to the piglets and influence piglets growth. Endometritis in the suckling phase can be a reason for reproductive failures in the following insemination – the authors also measured that and present these data in the results. Mastitis and Endometritis can falsify the results of this study. Moreover, blood markers for inflammation were measured – inflammatory processes like mastitis and endometritis will cover effects of given NSAID and SAID.

 L 127ff: the information about the parity of the 20 sows used for blood samples is missing. This information should be added. furthermore, what about statistical power? 20 sows per group seemed to be very low.

L 135: Due to crossfostering, I would suggest to present individual piglets weight and average daily weight gain than litter weights.

L 137ff: this paragraph is difficult to understand. Please rewrite it. Additionally, feed intake should not be “assumed” as written in L 141, feed intake should be really measured. In L 103 you wrote that sows stand on a fully slatted plastic floor – for sure you will not observe any food wastage as written in L 140. If feed intake if the feed intake was not measured properly, then these results should not be included in the study.

L 144ff: The use of PGFM and haptoglobin as markers for stress and inflammation is questionable. Haptoglobin is primary an indicator for hemolysis and the use of other acute-phase- proteins like C-reactive protein or serum-amyloid-A would be better indicators for inflammation. It is also not common to use PGFM as an inflammation marker, since PGFM is primary responsible for luteolysis. To determine C-reactive protein and serum-amyloid-A would enhance this part.

Statistics:

Was a power test analysis carried out to determine the minimum number of observations to correctly reject the null-hypotheses if this is false? If yes, the power test analysis should be described in detail. If no, I strongly suggest do analyse statistical power before starting an animal experiment because the used n/group and n/parity seemed very low to reach statistical power of 80%. Without a power test analysis, I cannot trust your results.

Results:

L 184: “The” should be There I guess.

L 194: instead of “little impact” I would rather say “no impact” because P-values are greater than 0.1

L 198: same as in L 194.

L 222ff: I cannot trust your results regarding feed intake because feed intake was not measured quantitatively but assumed.

L 232ff: here you presented data collected after weaning. In Materials and Methods it´s not described how animals were treated and handled after weaning, therefore it is hard to follow and many questions arises e.g. any hormonal treatment and other management procedures, housing. It is nearly impossible to interpret this data without having this information.

L 234: there are many reasons for low farrowing rates. Have any diagnostic procedure carried out to exclude pathogens (PRRSV, PCV2, Leptospira, Chlamydia, Parvovirus,…) being responsible for that beside the treatment?

Discussion:

In general, the discussion is rather superficial and some important points are not discussed or addressed:

L 243ff: Beside NSAIDs, there are many other reasons for stillborn piglets. These should be included and discussed in a broad context.

L 255: …..possible effects on the unborn child. – Reference is missing at the end of this sentence

L 256: …patency is maintained by prostaglandins. – Reference is missing at the end of this sentence

L 302: there are many other reasons for a poor reproductive performance which are not really discussed

Additionally, potential limitations of this study should be discussed.

Conclusion:

In the overall context of this study, the conclusion is rather speculative. Additionally it is written that dexamethasone improved sow-wellbeing although any information about this measurements and results are missing. This section should be revised.

L 397: Reference 27 is missing 

Author Response

Reviewer 1

The present study investigated the questions if meloxicam and dexamethasone given to sows prior to birth enhance piglets health and survial rate.

Introduction:

Overall, the introduction is focusing on elder references and some current references are missing. This should be added.

The references cited for anti-inflammatory administration of sows are from 2014 (12), 2016 (13) and 2020 (14). We have only included these three references as they related directly to the hypothesis, and we were intent on keeping the introduction brief.

Line (L) 45-48: The first sentence is difficult to understand and makes no sense in the overall context of the introduction. This sentence should be rephrased or rewritten.

We agree and this sentence has been deleted and its’ intent explained in the, now, second sentence

L 76 ff: The content of the stated aims of this study does not match the hypotheses that have been made. Aims are determining the effectiveness of Meloxicam and dexamethasone on farrowing and lactation performance of the sows and the hypothesis are focusing on sow´s comfort and reduced inflammatory processes and on piglets survival and growth. The authors should write this more precisely and decide on a focus.

We have added some text that hopefully addresses the reviewers concern.

L 80: please specify the term “reduce inflammation”, which inflammation is meant and where?

This sentence has been modified to clarify our meaning.

L 79ff: the authors hypothesied that the treatment improve sow comfort. There is no information in M+M how “sow comfort” was measured nor is there any information in the results section. This should be added, otherwise it must be assumed that some of the hypotheses have not been processed.

The reviewer is correct in that the concept of sow comfort is an assumption. We have modified the text to make this clearer

Materials and Methods:

Some additional information about the overall health status (e.g.PRRSV status) of the sow herd and vaccination programs (PRRSV, PCV2, Parvovirus, Erysipelas,…) would be very interesting. This information should be added.

Australia is PRRSv-free but other standard vaccines are now listed. Further, at no time were clinical signs observed that were consistent with any of these diseases.

L 123ff: A total of 13 sows developed mastitis or endometritis. Is there any information to which parity these sows belonged to? I would strongly recommand to exclude these sows from the study, as it is known that mastitis reduces the supply of milk/colostrum to the piglets and therefore effect their postnatal development. Additionally, antibiotics can be transmitted through milk to the piglets and influence piglets growth. Endometritis in the suckling phase can be a reason for reproductive failures in the following insemination – the authors also measured that and present these data in the results. Mastitis and Endometritis can falsify the results of this study. Moreover, blood markers for inflammation were measured – inflammatory processes like mastitis and endometritis will cover effects of given NSAID and SAID.

We thank the reviewer for this suggestion. We have now removed all sows treated for mastitis, re-analysed the data and updated the results. Whilst means and p-values changed slightly, there was no change in the outcomes of the experiment.

 L 127ff: the information about the parity of the 20 sows used for blood samples is missing. This information should be added. furthermore, what about statistical power? 20 sows per group seemed to be very low.

The number of sows in each parity range has been added to the text. We disagree with the comment made about low sample size. There are many examples of published work with smaller sample sizes than n=20

Verheyen AJM, Maes DGD, Mateusen B, Deprez P, Janssens GPJ, Lange Ld and Counotte G 2007. Serum biochemical reference values for gestating and lactating sows. The Veterinary Journal 174, 92-98.

L 135: Due to crossfostering, I would suggest to present individual piglets weight and average daily weight gain than litter weights.

We disagree as any effect would be due to the sow and the performance of her litter would therefore be sufficient for this purpose and not affected by cross fostering.

L 137ff: this paragraph is difficult to understand. Please rewrite it. Additionally, feed intake should not be “assumed” as written in L 141, feed intake should be really measured. In L 103 you wrote that sows stand on a fully slatted plastic floor – for sure you will not observe any food wastage as written in L 140. If feed intake if the feed intake was not measured properly, then these results should not be included in the study.

We thank the reviewer for this comment and we have now reworded to improve clarity.

Sows were fed twice daily at 0700 h and 1600 h by hand using a feed cart that had been converted to scales. Due to the design of the feeders (Wet/dry sow feeder, Crystal Spring Ste. Agathe, Canada), any feed wastage was retained in the bowl below the hopper, and before each feeding event, this wastage was returned to the hopper. Each feeder was filled to a standardised volume (marked by a lip at the top of the hopper), and so the feed delivered on each day was summed to give a daily feed intake for each sow.

We have inserted a photo of the feeders used so the reviewer can visualise how the wastage was captured.

L 144ff: The use of PGFM and haptoglobin as markers for stress and inflammation is questionable. Haptoglobin is primary an indicator for hemolysis and the use of other acute-phase- proteins like C-reactive protein or serum-amyloid-A would be better indicators for inflammation. It is also not common to use PGFM as an inflammation marker, since PGFM is primary responsible for luteolysis. To determine C-reactive protein and serum-amyloid-A would enhance this part.

We disagree the reviewer about the utility of PGFM and haptoglobin measurement. We have included why these measures were collected in the discussion:

Circulating PGFM levels reflect PGF2α biosynthesis and was used as a measure of prostaglandin-mediated inflammation [21]. However, a failure to detect a difference may reflect the short (15 min) half-life of PGFM and the need for more intensive sampling. Prostaglandin F2α is integral to the inflammatory process and inflammation also induces acute phase proteins including haptoglobin and C-reactive protein (CRP). Haptoglobin is commonly cited as being elevated during the transition from gestation to lactation in dairy cows [22], and anti-inflammatory agents have been shown to be successful in preventing the elevation of this APP [23]. Interestingly, while serum levels of haptoglobin and CRP have been shown to have minimal utility as an indicator of peripartal disease (Stiehler et al. 2016), haptoglobin is transferred to piglets via colostrum and colostrum concentrations were negatively associated with litter growth rate (Hiss-Pesch et al. (2011). Similarly, but to a lesser extent, piglet serum haptoglobin concentrations were negatively associated with growth (Hiss-Pesch et al. 2011).

Statistics:

Was a power test analysis carried out to determine the minimum number of observations to correctly reject the null-hypotheses if this is false? If yes, the power test analysis should be described in detail. If no, I strongly suggest do analyse statistical power before starting an animal experiment because the used n/group and n/parity seemed very low to reach statistical power of 80%. Without a power test analysis, I cannot trust your results.

 A power analyses was conducted prior to conducting the experiment using data obtained previously from the experimental site. Our main measure was day 21 litter weight. The previous dataset contained 989 litters, a mean d21 litter weight of 59.1, standard deviation of 14.7, standard error or 0.47 and variance of 0.25. Using this information, with n=50 sows per treatment we were able to detect a difference in day 21 litter weight of 5.7kg. Given the treatments only differed from the control by 1kg, these results are trustworthy.

In all our previous publications we have never been asked to include the power calculation in the methodology section and so feel it should not be included in this revision.

Results:

L 184: “The” should be There I guess.

Correct

L 194: instead of “little impact” I would rather say “no impact” because P-values are greater than 0.1.

We agree and change has been made.

L 198: same as in L 194.

We agree and change has been made.

L 222ff: I cannot trust your results regarding feed intake because feed intake was not measured quantitatively but assumed.

The authors have addressed this concern above.

L 232ff: here you presented data collected after weaning. In Materials and Methods it´s not described how animals were treated and handled after weaning, therefore it is hard to follow and many questions arises e.g. any hormonal treatment and other management procedures, housing. It is nearly impossible to interpret this data without having this information.

Post weaning information has been added to the text.

At weaning, all sows were relocated to a breeding barn and housed in groups of 40. Starting on day three post weaning, sows were run in front of mature boars for oestrus detection. At detection and again 24 h later, sows were artificially inseminated with 3 x 103 sperm in 80 mL diluent supplied by a commercial boar stud.

L 234: there are many reasons for low farrowing rates. Have any diagnostic procedure carried out to exclude pathogens (PRRSV, PCV2, Leptospira, Chlamydia, Parvovirus,…) being responsible for that beside the treatment?

We have reassessed the value of this result in the publication. After removing sows medicated for mastitis, we feel the sample size for farrowing rate is too low.

Discussion:

In general, the discussion is rather superficial and some important points are not discussed or addressed:

L 243ff: Beside NSAIDs, there are many other reasons for stillborn piglets. These should be included and discussed in a broad context.

We are aware that there are many reasons for stillbirths. However, in this case, it is associated with the NSAID treatment only, so a fuller discussion of stillbirth aetiology is not necessary.

L 255: …..possible effects on the unborn child. – Reference is missing at the end of this sentence

This is Antonucci et al (18). The text has been rewritten to clarify this.

L 256: …patency is maintained by prostaglandins. – Reference is missing at the end of this sentence

See above

L 302: there are many other reasons for a poor reproductive performance which are not really discussed Agreed, but in the context of the present study such reasons are not relevant.

Additionally, potential limitations of this study should be discussed.

We have discussed that timing of blood sampling for inflammatory markers may have been too late. We have also now included a section on how the experiment could have been strengthened by behavioural analyses around parturition.

Future work in this area should include behavioural observations on both the sow and piglets to confirm whether farrowing duration, and piglet vigour are impacted by anti-inflammatory administration.

Conclusion:

In the overall context of this study, the conclusion is rather speculative. Additionally it is written that dexamethasone improved sow-wellbeing although any information about this measurements and results are missing. This section should be revised.

We disagree as all conclusions are based on results obtained.

L 397: Reference 27 is missing

Sorry, our mistake, it was not in the text

Reviewer 2 Report

The article is well written amd from the experimental point of view, the study is well designed. The results and the conclusions can be fully underlined.

However, especially readers in Europe, where there is an ongoing vivid discussion about the wellbeing of food animals, will question, whether it is ethically justified to use any medicament in a natural process like farrowing to improve animal performance. In the text, it is said that "...anti-inflammatory administartion to sows prior to farrowing would (may) reduce pain and inflammation, thereby increasing piglet survival and growth."

This sentence is kind of an ethical justification, since animals in human care should be protected against pain (here of the sow), and unnecessary deaths (here of the piglets). But yet, the title of the paper suggests that the anti-inflammatory administartion to sows prior to farrowing is aimed at improving "lactation piglet performance".

I strongly recommend to reword the argumantation, why the authors have looked into the perceived improvemet of the animals wellbeing by using drugs prior to a natural biological process that happens in nature since uncountable millenniums without the hep of drugs. If the study was carried out since maybe farmers do believe that it would be better for the performance of sow and piglets, if..., and if the authors wanted to demonstarte that this is not a good idea, then I would say this in a friendly tone. But, as I said, I do not know the basic idea of the authors to test the effect of anti-inflammatory adminsitartion to sows prior to farrowing....

Author Response

Reviewer 2

The article is well written amd from the experimental point of view, the study is well designed. The results and the conclusions can be fully underlined.

However, especially readers in Europe, where there is an ongoing vivid discussion about the wellbeing of food animals, will question, whether it is ethically justified to use any medicament in a natural process like farrowing to improve animal performance. In the text, it is said that "...anti-inflammatory administartion to sows prior to farrowing would (may) reduce pain and inflammation, thereby increasing piglet survival and growth."

This sentence is kind of an ethical justification, since animals in human care should be protected against pain (here of the sow), and unnecessary deaths (here of the piglets). But yet, the title of the paper suggests that the anti-inflammatory administartion to sows prior to farrowing is aimed at improving "lactation piglet performance".

I strongly recommend to reword the argumantation, why the authors have looked into the perceived improvemet of the animals wellbeing by using drugs prior to a natural biological process that happens in nature since uncountable millenniums without the hep of drugs. If the study was carried out since maybe farmers do believe that it would be better for the performance of sow and piglets, if..., and if the authors wanted to demonstarte that this is not a good idea, then I would say this in a friendly tone. But, as I said, I do not know the basic idea of the authors to test the effect of anti-inflammatory adminsitartion to sows prior to farrowing....

We do not disagree with the reviewer and, unless justified by a reduction in stillbirths etc., our recommendation would be “hands off”. Yes, over unknown millennia sows have been farrowing without human intervention. However, in the current era of hyperprolific sows this is more difficult to justify. If we insist on sows delivering ever more pigs (which we think is the real ethical problem), then we need interventions to minimise the inevitable cost to the piglets. As for the title stating our investigation targets lactation piglet performance, the performance alluded to is survival and growth, both of which will be impacted by sow farrowing performance.